# Motives of Student Teachers in Academic Teacher Education for Secondary Education: Research in Flanders (Belgium) on the Motivation to Become and to Remain a Teacher

Katho Pauwels, Helena Van Loon, Els Tanghe and Wouter Schelfhout *

Antwerp School of Education, Faculty of Social Sciences, University of Antwerp, 2000 Antwerpen, Belgium
* Correspondence: wouter.schelfhout@uantwerpen.be

**Abstract:** The increasing shortage of teachers in secondary education, in Flanders (Belgium) as well as worldwide, is partly due to an insufficient inflow of new teachers. This stimulates research on the motivation of students to become and remain teachers. Our research focuses on the motives of student teachers in academic teacher education for secondary education to become a graduated teacher and how this evolves during teacher education, taking into account different profiles of students. Surveys, with closed and open questions based on our theoretical framework, were conducted at different points in time during teacher education as a basis for mixed method analyses. Compared to the period between the start and the end of the teacher education courses, it appeared that graduated student teachers attached more importance to the motives 'subject orientation' and 'educational orientation'. The research shows a shift to a more realistic educational orientation among graduated students. 'Work dynamic' aspects as a motive to work in education were generally considered equally important by women and men but combination possibilities were more important for men. Specifically related to students with a 'teacher in training' [TiT] statute, graduated TiT-students considered the motive 'idealism' more important than non-TiT-students. Related to working students, 'work perspective' appeared to be more important compared to non-working students. These insights into the evolving motives of student teachers have a number of possible implications for the organization of teacher education, which can also contribute to reducing the teacher shortage. Teacher education can put extra emphasis on positive and qualitative hands-on experiences by performing a well-considered selection of internship schools and mentors. At the same time, teacher education can offer even more space and time for student teachers to dedicate to internships, by keeping other activities such as deadlines and teaching moments during internships to a minimum.

**Keywords:** teacher education; student teacher motivation; teacher shortage

## 1. Introduction

The popularity of teaching has been under pressure in many industrialized countries worldwide for several decades, which is also the case in Belgium, leading to a teacher shortage [1]. The media emphasizes this social problem because school leaders do not find enough teachers [2,3]. Both the demand and supply side of education influence the teacher shortage. On the demand side, the rising number of pupils create a need for more teachers. A research in Flanders (part of Belgium) shows that the demand for education, for example in the age group 12–18, in the school year 2025–2026 is expected to be on average 16.8% higher than in 2015–2016 [4]. On the supply side, the teacher shortage on the one hand is due to insufficient teacher inflow. For example, the number of enrolments for teacher education is declining [5,6]. On the other hand, the teacher shortage on the supply side can be explained by a large outflow of teachers. Factors such as natural outflow, high work pressure, little future perspective, emotionally and physically demanding job and flat career play a role [7–11]. The first five years are crucial for beginning teachers, as they are decisive

for the further development of a career in education [12]. In Flanders up to 44% of starting teachers leave secondary education within five years [13].

Research into the motives of student teachers to become and remain teachers is an added value to gain in-depth insights into the teacher shortage [12]. The present study uses data from surveys on the motives of student teachers at the University of Antwerp (Belgium). The corresponding general research question is: What motivates student teachers to become teachers (within the teacher education program of the University of Antwerp)? An important development that is occurring in teacher education, in Flanders but also in many other countries worldwide where there is a clear teacher shortage, is that more and more students are combining teacher education with actual teaching [14,15]. The practical component of the training is then fully implemented as a teacher in a particular school, with both the school and teacher education providing coaching. This context may have advantages, but also pitfalls related to the degree to which a student teacher will still be open to learning from teacher education [16]. Since this is an increasingly large group of students, it is important to check whether the motives of these students differ from the other students and whether they evolve differently throughout their education.

## 2. Theoretical Framework

### 2.1. Motives of Student Teachers

To address the teacher shortage, it is crucial to know what motivates student teachers to really become teachers. First of all the existing literature shows that teaching is often associated with the motivation to educate people, share knowledge and values and help a community move forward [17]. Several studies on student teachers confirm this feeling of idealism and pupil orientation [18–20]. These studies conclude that societal considerations such as making a social contribution, shaping the future, a sense of calling, fulfilling a dream and working with children/adolescents are among the most important motivations for choosing the teaching profession [18,20,21].

In addition, aspects such as subject orientation and educational orientation play an important role for choosing the teacher profession. Pedagogical considerations such as the value and activity of teaching, commitment to the subject and knowledge transfer are considered to be important [22–24]. Work dynamic is also a motive. Jarvis and Woodrow [25] find that students' motivation is often related to the expectation that being a teacher will be a challenging and fulfilling career. Students who choose teacher education for the reasons mentioned above consider these motives to be more important than combination possibilities and work perspectives, such as plenty of free time, long holidays, a reasonable salary, job security, the certainty of a pension and the association of having to work relatively little [24,26,27]. Combination possibilities, in turn, appear to be a primary motive for women with children. They may opt for teacher education in order to cope with changing family circumstances [19]. Other students follow a teacher education to have more options on the labor market. These students have often not yet decided whether they actually want to work as a teacher after teacher education [28]. Another reason to become a teacher is students' own experience in teaching. Teachers do influence the motivation of students to enter the teaching profession themselves. Both "good" and "bad" teachers influence that motivation. "Good teachers" ensure that student teachers mirror them as closely as possible, while "bad teachers" serve as examples of how not to do it [29].

In any case it is important to take into account the context in which the student teacher finds himself because it leads to the formation of the teacher identity. For example, the political, social and cultural context in which students find themselves influence the motivation to enter teacher education. A student might be motivated to become a teacher because a family member recommends it. The social and political context also plays a role. Indeed, the teaching profession is said to bring job security and status [29]. Nevertheless, status does not always influence motivation in a positive way. Although prestige and social status are considered to be characteristics of the teaching profession [6], the international TALIS study [11] shows that teachers often struggle with the feeling that their work is

undervalued in society. This is because the teaching profession is commented on and evaluated more than any other profession by students, parents, school leaders, the general public, policy makers and international organizations. This is problematic because a sense of appreciation, both by school actors and society, is important for retaining teachers and persuading students to choose the teaching profession [30].

A study about the evolution of the motivation of students from teacher education at Ghent University (Flanders, Belgium) shows that the students felt more strongly committed to the teaching profession after teacher education [31]. Self-determination theory makes it clear that commitment is one of the three essential needs that must be satisfied in order to experience some form of motivation [32]. In the absence of commitment, which is defined as the degree of psychological attachment to the teaching profession [33], there will be less motivation to actually work as a teacher [34,35]. As involvement has become stronger, the motivation to become a teacher will probably also be stronger. The same research found that the most important motivators to become a teacher are social utility values, student intrinsic values, perceived teaching skills, and previous learning and teaching experience. This research defines social utility values as shaping the future of children, making a social contribution and working with children and young people [31]. A second observation is the fact that material motives or combination possibilities and work perspective are considered less important [24,26,27]. The study investigated the evolution of student teacher motivation by measuring their motivation at both the beginning and the end of the program. Students generally scored high on the above factors, both at the beginning and at the end of the teacher education. Motivation in general hardly changed throughout the program. Social utility values are the exception here. This motive strengthened during the training. It means that students discover the social added value of being a teacher during the training [31].

It is important to investigate the motives of side entrants specifically because in many countries with a teacher shortage, initiatives are taken to attract people with work experience into the education sector [36]. Within teacher education in Flanders (Belgium), there is a large group of side entrants. The Flemish Ministry of Education [37] defines side entrants as teachers who decide to switch to a job in education after a previous work experience. Applied to this research, side entrants include both student teachers with a status of teacher-in-training (TiT-student) and working students. In Flemish education, a TiT-student is someone who is already practicing a teaching profession, but who is at the same time taking teacher education courses. TiT-students follow the theoretical subjects of teacher education together with the regular students, but for the practical parts they are (partially) valorized. TiT-students can carry out internships as part of their job but are still supported by the teacher education department. Working students are students who combine working and studying. Within this status it is not required to have a job within education. Working students can also be employed in other sectors. At the University of Ghent [38], the motivation of side entrants specifically for adult teacher education was investigated. This research concludes that side entrants consider the motive pupil orientation to be especially important since they want to work with adolescents. Educational orientation also appears to dominate within this target group. Motives such as work perspective and idealism score well, but slightly less high than education and pupil orientation [38].

The above motives of student teachers come together as a synthesis in the study by Meeus, Baeten and Coertjens [39]. This study examines motives of future teachers who start a one-year teacher education program at a Flemish university (part of Belgium). The obtained model comprises seven important motives that play a role in the choice for teaching. The present study thus uses this model as a starting point for further analyses for various reasons. Firstly, the motives were developed on the basis of student teachers following a one-year teacher education program at a Flemish university. Due to the very similar target group with the current research, the measurement of these motives is therefore largely applicable. Secondly, the motives are more detailed compared to other

theoretical frameworks [40,41]. This constitutes an added value with a view to policy recommendations for teacher education programs.

The seven motives that play a role in the choice for education are the following [39]:

(a)  Subject orientation: wanting to motivate pupils for the subject, wanting to teach pupils new subject content, wanting to help pupils understand the subject better, wanting to make the subject exciting;

(b)  Work perspective: job security, chance of work, not seeing any other work opportunities, a logical consequence of the choice of study, job reorientation due to dismissal, physical complaints or wanting another profession;

(c)  Work dynamic: wanting work with a lot of variation, wanting a dynamic job, preferring a lot of autonomy, liking challenges;

(d)  Educational orientation: like to be in front of an audience, like to guide students, like to talk, like to teach, like to develop material;

(e)  Pupil orientation: wanting to help pupils, offer pupils opportunities, prepare pupils for further studies or the labor market, show pupils what they are capable of;

(f)  Idealism: wanting to make a social contribution, help improve the world a little, help shape the next generation, stir pupils' idealism;

(g)  Combination possibilities: wanting to work pleasant hours, work close to home, have a good balance with family life, have lots of holidays.

The extent to which certain motives manifest themselves is influenced by personal factors as well as the political, social and cultural context. The above characteristics form the basis for gaining a deeper insight into the motives of student teachers and alumni for choosing or continuing to choose the teaching profession [42]. This framework therefore forms the starting point for theoretically examining how the motives differ between student teachers and how they subsequently evolve.

### 2.2. Research Questions

Based on this theoretical framework, the following research questions are formulated: What motivates student teachers to become teachers? This research question can be divided into three sub-questions.

(a)  What is the motivation of students to become teachers at the start of teacher education?

(b)  How and to what extent does the motivation of students to become a teacher evolve during teacher education between the start and at the end when graduated?

(c)  What are possible differences between regular students, working students (students that combine another job with teacher education) and 'teachers-in-training' (students that combine teaching with teacher education)?

## 3. Materials and Method

### 3.1. Participants

This study used data from surveys on the motivation of students following a teacher education at the University of Antwerp during the 2020–2021 academic year. During the General Didactics lecture, the group of newly registered students was asked to fill out an online survey made in Qualtrics about their motivation to become a teacher. The content part of the written questionnaire was based on the validated scale of Meeus, Baeten and Coertjens [39]. In 2019–2020, the questionnaire was expanded to include a section on personal data and questions on the organization of the course. We tested and validated these additional questions by checking content and face validity [43]. The first version of these questions was based on literature review and then presented to a number of experts in the field of teacher education. Based on this feedback, the initial version was further adapted. For example, the personal questions in the first version were not completely geared to the various intake of students, the distinction could be better in terms of whether or not work experience and in which sector. Next, we checked whether student teachers and teacher educators, found the adapted questionnaire of sufficient quality in terms of

feasibility, readability, consistency of style and formatting, and the unambiguity and clarity of the language used. We did this by administering the test to a sample panel of 10 student teachers and by means of a focus discussion on these different aspects for each question, deleting questions, adjusting them and adding them where necessary [44].

Out of 315 registered students, 232 participated in the survey at the beginning of academic year 2020–2021, which resulted in a response rate of 73.7% for this measurement at the start of teacher education. In the period June to September 2021, the survey was conducted with graduated students of the educational master's program in academic year 2020–2021 with the aim of determining the potential change compared to the results of the starting students. Of the 100 graduating students, 49 participated in the survey. This resulted in a response rate of 49%.

### 3.2. Variables

The respondents indicated to what extent they attached importance on a number of statements concerning different aspects of motivation. The response categories of the statements were "very little important", "not very important", "somewhat important", "important" and "very important". The statements were grouped by theme and then combined to form the seven motives from the theoretical framework (see Table 1, for more examples see Appendix A).

**Table 1.** Variable statement.

| Variable | Statements |
|---|---|
| Subject orientation | I like to motivate students for my subject |
| Work perspective | I want to work in education because it offers great job security |
| Work dynamic | I want a job with a lot of variety |
| Educational orientation | I enjoy teaching |
| Pupil orientation | I want to help pupils in their choice of study or profession |
| Idealism | I want to improve education |
| Combination possibilities | I like to have good working hours |
| Gender | Gender is broken down into male, female and X. |
| Working student | Students who combine work and study. Within this statute, it is not a requirement to have a job in education. Working students can also be employed in other sectors. |
| TiT-student | Teacher-in-training: someone who is already practicing as a teacher, but who is simultaneously taking teacher education courses. TiT-students follow the theoretical subjects of teacher education together with regular students, but are (partially) valorized for the practical components. TiT-students can carry out internships as part of their job but are still supported in this by teacher education. |

### 3.3. Method

Each variable was asked by means of an item whereby the respondents (see Table 2) could answer on the following Likert scale: very little important (1), little important (2), somewhat important (3), important (4) and very important (5). A number of items were combined to form scales. In order to facilitate reporting, mean and standard deviation, together with Cronbach's alpha ($\alpha$) indicating reliability, are calculated for each scale (see Table 3).

**Table 2.** Respondents.

|  |  | Started Students | Graduated Students |
|---|---|---|---|
|  |  | %N | %N |
| Gender | Male | 33.6 | 28.6 |
|  | Female | 64.7 | 69.4 |
|  | X | 1.7 | 2.0 |
| TiT | TiT | 15.1 | 24.5 |
|  | Non-TiT | 84.9 | 75.5 |
| Working student | Working | 22.4 | 26.4 |
|  | Non-working | 77.6 | 73.5 |
| Regular student | Regular | 47.4 | 36.7 |
|  | Non-regular | 52.6 | 63.3 |

**Table 3.** Descriptive statistics (percentage distribution (%) for categorical variables).

|  | Started Students | | | Graduated Students | | |
|---|---|---|---|---|---|---|
|  | Mean | *S.D.* | $\alpha$ | Mean | *S.D.* | $\alpha$ |
| Subject orientation | 3.875 | 0.636 | 0.873 | 3.510 | 0.845 | 0.904 |
| Work perspective | 2.095 | 0.800 | 0.686 | 1.837 | 0.986 | 0.720 |
| Work dynamic | 3.487 | 0.727 | 0.772 | 3.510 | 0.845 | 0.823 |
| Educational orientation | 3.203 | 0.707 | 0.718 | 3.204 | 0.706 | 0.532 |
| Pupil orientation | 3.776 | 0.618 | 0.835 | 3.571 | 0.791 | 0.814 |
| Idealism | 3.440 | 0.814 | 0.844 | 3.388 | 1.017 | 0.870 |
| Combination possibilities | 2.987 | 0.860 | 0.751 | 2.918 | 1.057 | 0.744 |

*3.4. Data Analysis*

Mixed methods were used to answer the research questions. Quantitative data management in preparation and the analyses themselves were done in the statistical program "IBM SPSS Statistics 28". In order to find out whether the motives of students who had started their studies differed significantly from the motives of graduates, independent *t*-tests were conducted. The variables used have an ordinal measurement level and were approximately normally distributed. Since the sample was more than 100, conducting an independent *t*-test was sufficiently robust [45]. The *t*-tests were supplemented with qualitative data from open survey questions. One open answer could contain several aspects of the seven categories of Meeus, Baeten and Coertjens [39]: subject orientation, work perspective, work dynamic, educational orientation, student orientation, idealism, combination possibilities. During data processing, these answers were coded into quantifiable data and classified into the seven motives of the theoretical framework. Coding was done by two people, to reduce the chance of observer bias.

## 4. Results

*4.1. Motives for Started and Graduated Students*

Table 4 shows the results in terms of the difference in motivation between started and graduated students. The quantitative results were supplemented with qualitative data. In order to know to what extent a certain motive was considered important by student teachers, it was examined how often each motive was formulated and in what way. Figure 1 shows these qualitative results for the difference in motivation between started and graduated students.

**Table 4.** Independent *t*-test for difference in motivation between started students and graduated students (N = 281).

| | Started Students | | Graduated Students | | | | | |
|---|---|---|---|---|---|---|---|---|
| | Mean | *SD* | Mean | *SD* | *t* | *df* | Sig. | *d* |
| Subject orientation | 3.875 | 0.636 | 3.510 | 0.845 | 2.857 | 59.997 | 0.006 ** | 0.539 |
| Work perspective | 2.095 | 0.800 | 1.837 | 0.986 | 1.716 | 62.025 | 0.091 | 0.309 |
| Work dynamic | 3.487 | 0.727 | 3.510 | 0.845 | −0.197 | 279 | 0.844 | −0.031 |
| Educational orientation | 3.202 | 0.707 | 3.204 | 0.706 | −0.013 | 279 | 0.989 | −0.002 |
| Pupil orientation | 3.776 | 0.618 | 3.571 | 0.791 | 1.703 | 60.997 | 0.094 | 0.314 |
| Idealism | 3.440 | 0.814 | 3.388 | 1.017 | 0.335 | 61.646 | 0.739 | 0.061 |
| Combination possibilities | 2.987 | 0.860 | 2.918 | 1.057 | 0.426 | 62.106 | 0.672 | 0.077 |

** *p* < 0.01.

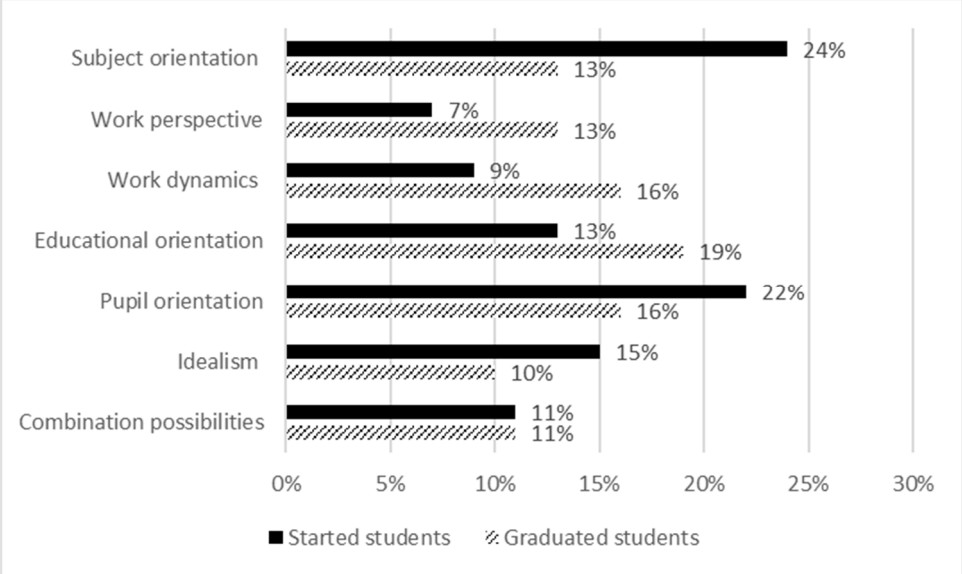

**Figure 1.** Motives for started students versus graduated students, n = 281.

The descriptive data showed that started students considered subject orientation and pupil orientation the most important motives (M = 3.875, SD = 0.636 and M = 3.776, SD 0.618). The qualitative results also indicated that subject orientation and pupil orientation are the most important reasons for choosing education. The quotes "pass on my passion for science and mathematics", "I want to teach pupils new knowledge" and "convey passion for the job" link to subject orientation. "Enjoy working with young people and preparing them for the future" refers to pupil orientation.

Graduated students also found subject orientation (M = 3.510, SD = 0.845) and pupil orientation (M = 3.571, SD = 0.791) to be the most important motives, but rated it as less important compared to the start of the academic year. The motives subject orientation and pupil orientation also remained the most important according to the qualitative data. Student teachers indicate this as follows: "I want to make a connection with young people by passing on new knowledge" which corresponds to pupil orientation or "I like to speak in front of a group" which corresponds to educational orientation.

In order to observe the evolution between started students and graduated students, the descriptive results were further explored by means of *t*-tests and qualitative data. The results of these analyses are presented schematically in Table 2.

The difference in mean subject orientation for started students (M = 3.875, SD = 0.636) and graduated students (M = 3.510, SD = 0.845) was significant: t (59.997) = 2.857, *p* < 0.01,

with a medium effect size (d = 0.539). These quantitative results are also confirmed by the qualitative findings. For example, a started student indicated that "encouraging students to like science" was an important reason for choosing education.

When examining at the quantitative difference in mean work perspective for started students (M = 2.095, SD = 0.800) and graduated students (M = 1.837, SD = 0.986), this difference was slightly significant at the exploratory level: t (62.025) = 1.716, *p* < 0.1, with a medium effect size (d = 0.309). Although the quantitative *t*-test showed a decrease, it does not mean that graduate students consider work perspective negligible. Indeed, the qualitative data showed that they did attach importance to it. A possible explanation could be that towards the end of the academic year, student teachers are looking for a job. For example, one graduate student indicated that "There is job security as a French teacher" was an important reason. Furthermore, the qualitative data showed that work perspective remained less important as a motive compared to other motives.

Additionally, the difference in mean pupil orientation for started students (M = 3.776, SD = 0.618) and graduates (M = 3.571, SD = 0.791) was slightly significant at the exploratory level: t (60.997) = 1.703, *p* < 0.1, with a medium effect size (d = 0.314). This decrease was also seen in the qualitative data. Although educational orientation was not found to be significant according to quantitative data, qualitative data showed an increase.

The results show a shift from a more idealistic pupil orientation among started students to a more realistic educational orientation among graduated students. Started students underline their motives with the following quotes: "To teach and motivate students, to give students opportunities". Statements by graduated students that are more in line with an educational orientation are "I want a social job where you come into contact with many people, I like standing in front of a class, I like speaking in front of a group, teaching provides energy".

### 4.2. Motives for Started and Graduated Students by Gender

In the group of started students, 33.6% of the respondents were male, 64.7% female and 1.7% X (see Table 2). In the group of graduates, 28.6% was male, 69.4% female and 2% X. There was thus a majority of women in both respondent groups. The sample for category X is too small to perform statistically representative calculations. For that reason, the category was omitted.

When examining the started students, the extent to which work dynamic was a significant motive for starting as a teacher was significantly different for men (see Table 5, M = 3.772, SD = 0.648) and women (M = 3.975, SD = 0.655) at t (230) = 2.244, *p* < 0.05, with a medium effect size (d = 0.312). According to the qualitative data, deduced from Figure 2, started men and women considered work dynamic, as a reason for choosing education, equally important. Quotes such as "an active job that keeps challenging you" and "not sitting still all day" illustrate this.

**Table 5.** Independent *t*-test for difference in motivation between male and female for started students (n = 228) and graduated students (n = 48).

| | Started Students | | | | | | | | Graduated Students | | | | | | | |
| | Male | | Female | | | | | | Male | | Female | | | | | |
| | Mean | SD | Mean | SD | t | df | Sig. | d | Mean | SD | Mean | SD | t | df | Sig. | d |
|---|---|---|---|---|---|---|---|---|---|---|---|---|---|---|---|---|
| Subject orientation | 4.288 | 0.518 | 4.293 | 0.574 | 0.072 | 230 | 0.942 | 0.010 | 3.847 | 0.356 | 3.927 | 0.900 | 0.445 | 46.987 | 0.658 | 0.102 |
| Work perspective | 2.451 | 0.674 | 2.474 | 0.797 | 0.210 | 230 | 0.834 | 0.029 | 2.089 | 0.880 | 2.090 | 1.088 | −0.011 | 47 | 0.991 | −0.003 |
| Work dynamic | 3.772 | 0.648 | 3.975 | 0.655 | 2.244 | 230 | 0.026 * | 0.312 | 4.000 | 0.715 | 3.891 | 0.925 | −0.394 | 47 | 0.695 | −0.125 |
| Educational orientation | 3.646 | 0.609 | 3.636 | 0.675 | −0.108 | 230 | 0.914 | −0.015 | 3.857 | 0.589 | 3.531 | 0.676 | −1.578 | 47 | 0.121 | −0.499 |
| Pupil orientation | 4.126 | 0.556 | 4.241 | 0.584 | 1.443 | 230 | 0.150 | 0.201 | 4.011 | 0.450 | 4.000 | 0.847 | −0.089 | 42.954 | 0.929 | −0.022 |

**Table 5.** *Cont.*

|  | Started Students | | | | | | | | Graduated Students | | | | | | |
|  | **Male** | | **Female** | | | | | | **Male** | | **Female** | | | | | |
| Idealism | 3.862 | 0.682 | 3.801 | 0.820 | 0.107 | 230 | 0.577 | −0.078 | 3.957 | 0.598 | 3.610 | 1.120 | −1.418 | 42.826 | 0.163 | −0.350 |
| Combination possibilities | 3.385 | 0.833 | 3.409 | 0.796 | 0.218 | 230 | 0.828 | 0.030 | 3.630 | 0.764 | 3.050 | 1.103 | −2.079 | 34.516 | 0.045 * | −0.563 |

Note: * = *p* < 0.05.

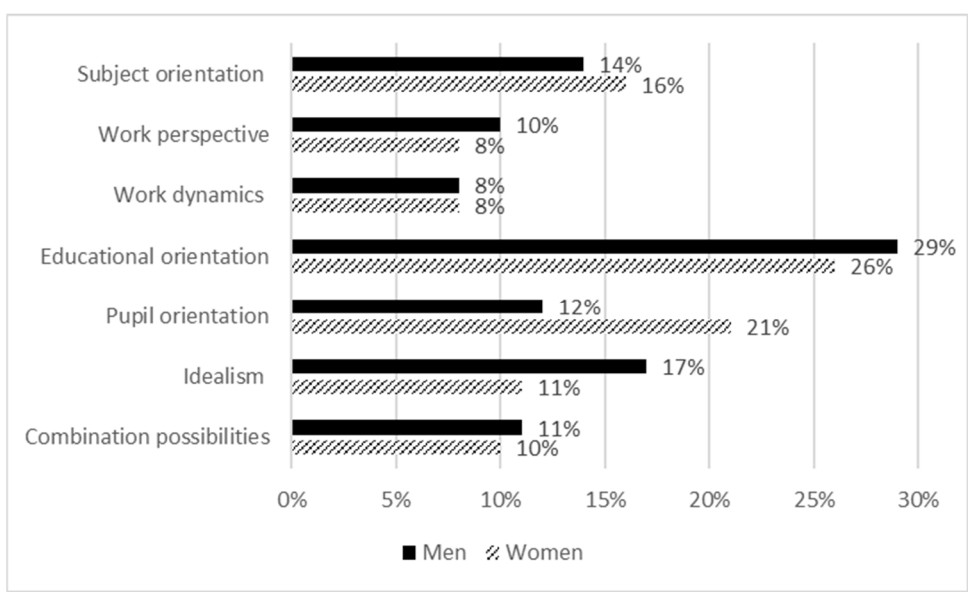

**Figure 2.** Motives of started male versus female (n = 228).

When we examine the graduated students, it turned out that the motive combination possibilities for men (M = 3.630, SD = 0.764) and women (M = 3.050, SD = 1.103) were significantly different: t (34.516) = −2.079, *p* < 0.05, with a medium effect size (d = −0.563). This was also demonstrated by the following statement from a man: "I can plan my work and private life well by myself".

When qualitative data (Figures 2 and 3) are compared with men who graduated, the following evolutions stand out. Work dynamic was more important for graduated men than for men who had started their studies. Work perspective was also seen as a more important reason by graduated men compared to started men. This can be explained by the hands-on experience that graduates have had. After teacher education, students have a better idea of what the job entails. This may cause the dynamic element, as well as the will for security in the teaching profession to become more important. When the qualitative data of the women were subsequently compared, a similar evolution emerged. Work dynamic also became more important as teacher education progressed. The same explanation can therefore be given as for the male respondents.

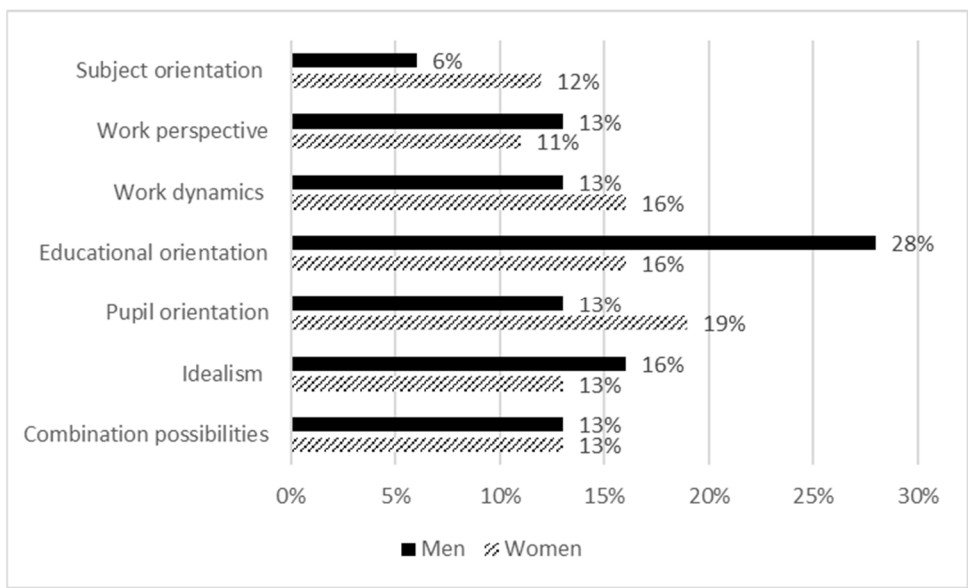

**Figure 3.** Motives of graduated male versus female (n = 48).

*4.3. Motives of Started and Graduated TiT-Students*

The descriptive data in Table 2 show that of the students who had started their studies, 15.1% were TiT-students and 84.9% were non-TiT-students. Of the group of graduates, 24.5% had the TiT-status and 75.5% did not. As described before, in Flemish education a TiT-student is someone who is already practicing a teaching profession and who is simultaneously following a teacher education program.

When we are examining the graduated students only the difference in mean idealism for TiT-students (see Table 6, M = 4.117, SD = 0.522) and non-TiT-students (M = 3.573, SD = 1.091) was significant: t (39.843) = −2.320, $p < 0.05$, with a medum effect size (d = −0.550). In other words, the mean idealism is higher among TiT-students compared to non-TiT-students. The qualitative data, shown in Figure 4, also confirm this finding. Some TiT-students, for example, made the following statements: "preparing students for the world, I want to help young people find a way in life", "as a teacher, I want to make a difference for pupils" and "I want to contribute to young people's lives in a pleasant way".

**Table 6.** Independent *t*-test for difference in motivation between TiT-students and non-TiT-students for started (n = 232) and graduated (n = 49).

| | Started Students | | | | | | | | Graduated Students | | | | | | | |
| | TiT-Student | | Non-TiT-Student | | | | | | TiT-Student | | Non-TiT-Student | | | | | |
| | Mean | SD | Mean | SD | t | df | Sig. | d | Mean | SD | Mean | SD | t | df | Sig. | d |
|---|---|---|---|---|---|---|---|---|---|---|---|---|---|---|---|---|
| Subject orientation | 4.449 | 0.479 | 4.263 | 0.564 | −1.834 | 230 | 0.068 | −0.336 | 3.905 | 0.410 | 3.904 | 0.868 | −0.005 | 47 | 0.996 | −0.002 |
| Work perspective | 2.621 | 0.723 | 2.439 | 0.761 | −1.316 | 230 | 0.189 | −0.241 | 2.208 | 1.200 | 2.047 | 0.977 | −0.470 | 47 | 0.641 | −0.156 |
| Work dynamic | 3.943 | 0.587 | 3.901 | 0.671 | −0.350 | 230 | 0.727 | −0.064 | 3.817 | 0.562 | 3.957 | 0.945 | 0.484 | 47 | 0.630 | 0.161 |
| Educational orientation | 3.823 | 0.649 | 3.607 | 0.649 | −1.813 | 230 | 0.071 | −0.333 | 3.750 | 0.724 | 3.584 | 0.647 | −0.751 | 47 | 0.456 | −0.250 |
| Pupil orientation | 4.329 | 0.498 | 4.180 | 0.587 | −1.407 | 230 | 0.161 | −0.258 | 4.208 | 0.370 | 3.932 | 0.831 | −1.109 | 47 | 0.273 | −0.368 |
| Idealism | 3.806 | 0.835 | 3.824 | 0.767 | 0.131 | 230 | 0.896 | 0.024 | 4.117 | 0.522 | 3.573 | 1.091 | −2.320 | 39.843 | 0.026 * | −0.550 |
| Combination possibilities | 3.571 | 0.780 | 3.371 | 0.810 | −1.360 | 230 | 0.175 | −0.249 | 3.104 | 1.290 | 3.250 | 0.968 | 0.360 | 15.231 | 0.724 | 0.139 |

* $p < 0.05$.

When comparing the qualitative data (Figures 4 and 5) from started TiT-students with graduated TiT-students, the following differences stand out. Educational orientation was considered a more important reason by graduated TiT-students compared to started TiT-students. Combination possibilities ("work-live balance, lots of holidays, nice working hours") were also considered more important by graduated TiT-students compared to started TiT-students. TiT-students follow teacher education in combination with the busy job as a teacher. They may experience this as too heavy which may cause them to start seeing the importance of combination possibilities more and more. On the other hand, pupil orientation was less important among TiT-graduates compared to TiT-students who had just started their education. This result may be explained by the survival phase that beginning teachers go through. Teachers in this phase switch to autopilot and find little time to focus on things besides teaching, such as pupil orientation. Work perspective as a reason for choosing teaching also decreased in importance from started TiT-students to graduated TiT-students. A possible explanation for this may be that TiT-students already have a job and therefore experience less or no stress in finding one.

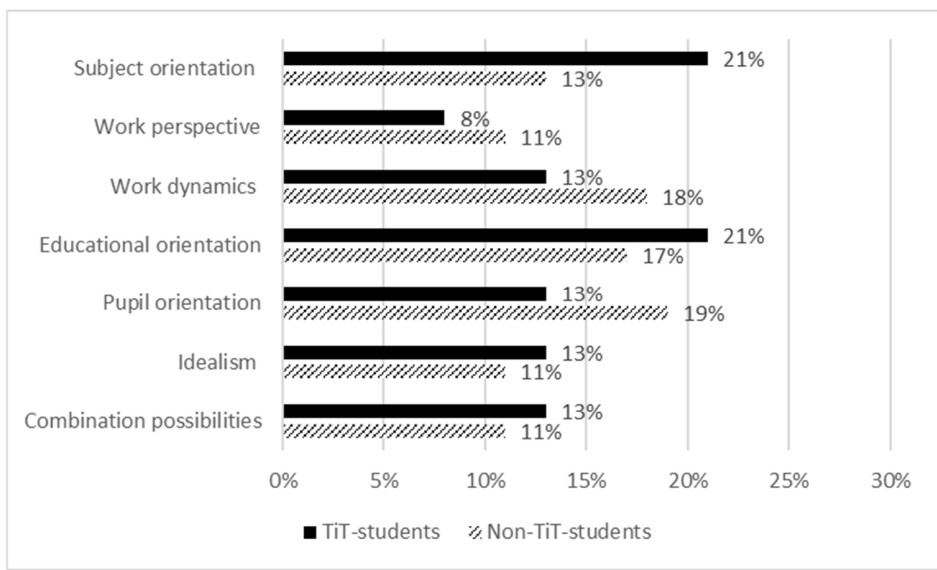

**Figure 4.** Motives of TiT-graduates versus non-TiT students (n = 49).

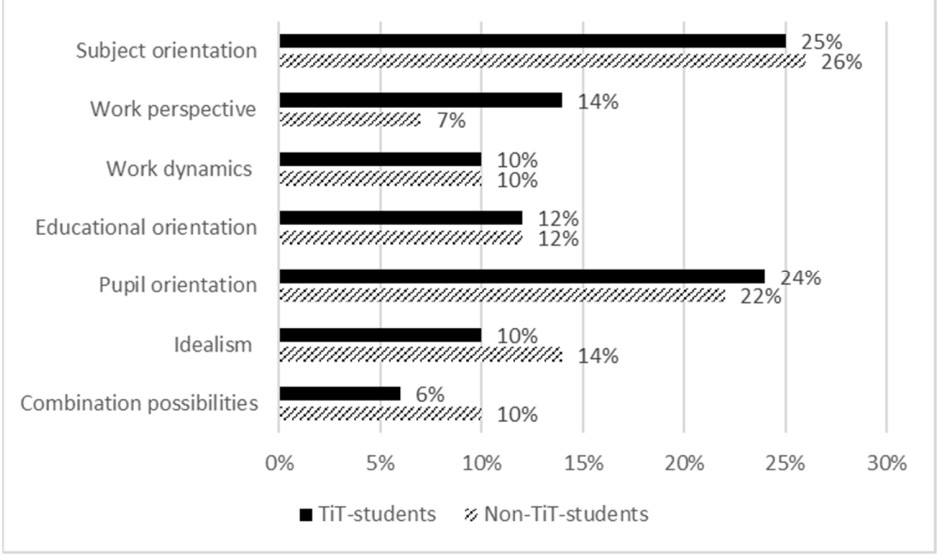

**Figure 5.** Motives for started TiT-student versus non-TiT-student (N = 232).

### 4.4. Motives for Started and Graduated Working Students

The descriptive data in Table 2 show that of the students who started their studies, 22.4% were working students and 77.6% were non-working students. Of the group of graduates in the same academic year, 26.4% had the status of working students and 73.5% did not.

Among the started students, the difference in mean work perspective for working students (see Table 7, M = 2.120, SD = 0.671) and non-working students (M = 2.567, SD = 0.752) was significant: t (230) = 3.860, $p < 0.001$, with a medium effect size (d = 0.608).

**Table 7.** Independent *t*-test for difference in motivation between working students and non-working students for started (n = 232) and graduated (n = 49).

| | Started Students | | | | | | | | Graduated Students | | | | | | | |
| | Working Students | | Non-Working Student | | | | | | Working Students | | Non-Working Students | | | | | |
| | Mean | SD | Mean | SD | t | df | Sig. | d | Mean | SD | Mean | SD | t | df | Sig. | d |
|---|---|---|---|---|---|---|---|---|---|---|---|---|---|---|---|---|
| Subject orientation | 4.236 | 0.543 | 4.307 | 0.559 | 0.811 | 230 | 0.418 | 0.128 | 3.429 | 1.063 | 4.075 | 0.571 | 2.088 | 14.577 | 0.055 | 0.888 |
| Work perspective | 2.120 | 0.671 | 2.567 | 0.752 | 3.860 | 230 | <0.001 *** | 0.608 | 1.885 | 0.911 | 2.160 | 1.064 | 0.828 | 47 | 0.412 | 0.268 |
| Work dynamic | 4.031 | 0.628 | 3.871 | 0.664 | −1.545 | 230 | 0.124 | −0.243 | 3.462 | 1.258 | 4.089 | 0.611 | 1.726 | 14.098 | 0.106 | 0.760 |
| Educational orientation | 3.550 | 0.622 | 3.666 | 0.660 | 1.126 | 230 | 0.261 | 0.177 | 3.308 | 0.724 | 3.739 | 0.610 | 2.079 | 47 | 0.043 * | 0.673 |
| Pupil orientation | 4.186 | 0.573 | 4.207 | 0.578 | 0.237 | 230 | 0.813 | 0.037 | 3.385 | 0.843 | 4.222 | 0.581 | 3.311 | 16.316 | 0.004 ** | 1.273 |
| Idealism | 3.831 | 0.688 | 3.819 | 0.801 | −0.097 | 230 | 0.923 | −0.015 | 3.139 | 1.387 | 3.911 | 0.752 | 1.910 | 14.628 | 0.076 | 0.809 |
| Combination possibilities | 3.313 | 0.856 | 3.426 | 0.793 | 0.896 | 230 | 0.371 | 0.141 | 2.904 | 1.197 | 3.326 | 0.976 | 1.259 | 47 | 0.214 | 0.407 |

\* $p < 0.05$; ** $p < 0.01$; *** $p < 0.001$.

Work perspective was considered a more important motive by non-working students compared to working students. The qualitative results, shown in Figure 6, also confirm this. A possible explanation for this is that working students already have a job and therefore experience less stress in trying to find one. It could be that job security in education is less important to this group of respondents.

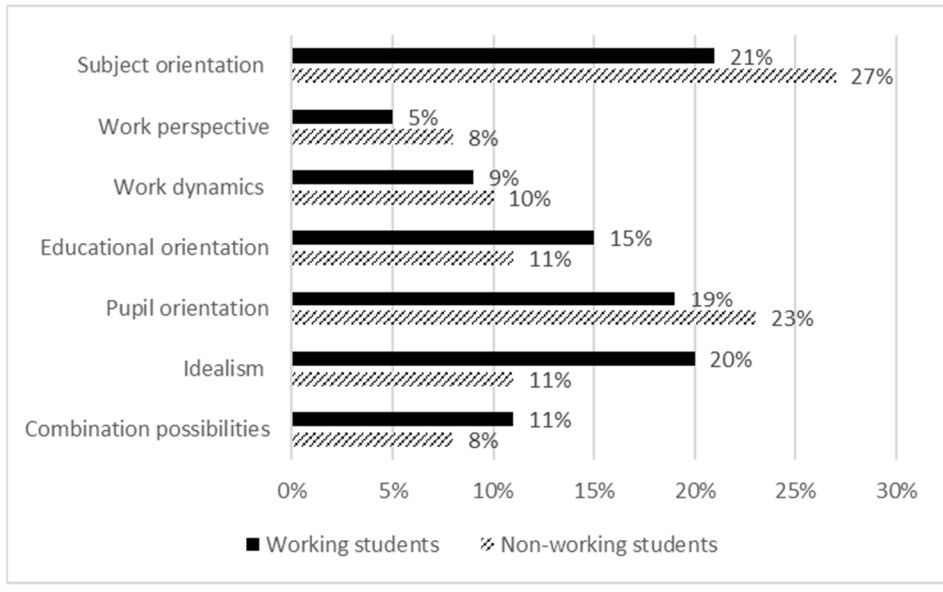

**Figure 6.** Motives of started working students versus non-working students (N = 232).

The difference in mean subject orientation for the graduates between working students (M = 3.429, SD = 1.063) and non-working students (M = 4.075, SD = 0.571) was slightly significant at the exploratory level: t (14.577) = 2.088, *p* < 0.1, with a medium effect size (d = 0.888). Based on the *t*-tests, subject orientation appeared to be more important for non-working students compared to working students. This was also observed in the qualitative data (see Figure 7). This finding goes against our expectations as working students already have experience in the field and can therefore share their expertise with students. For working students, from all their experience, subject orientation may be so evident that they no longer think about it in the first place.

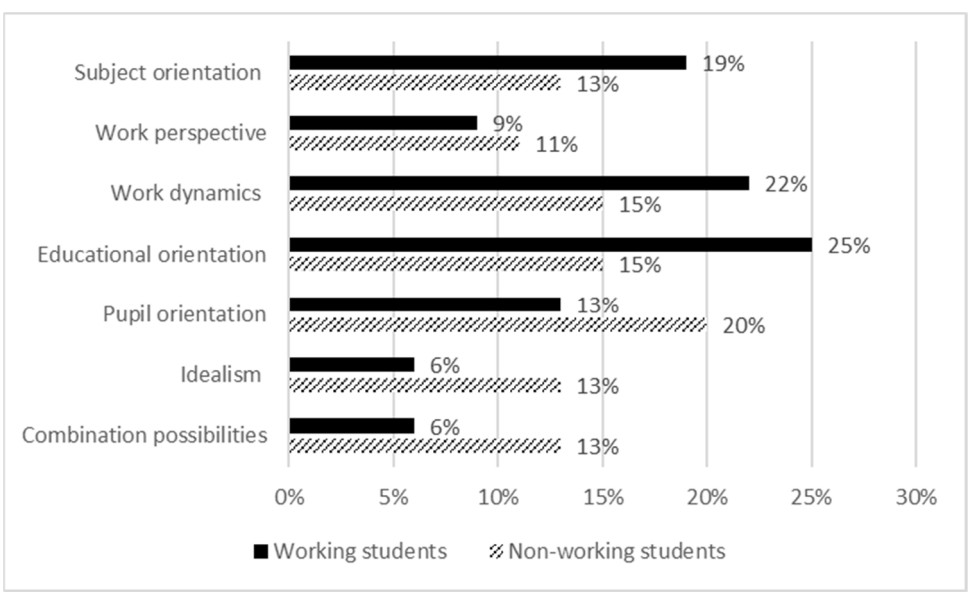

**Figure 7.** Motives of graduated working student versus non-working student (N = 49).

In the group of graduates, the difference in mean educational orientation for working students (M = 3.308, SD = 0.724) and non-working students (M = 3.739, SD = 0.610) was significant: t (47) = 2.079, *p* < 0.05, with a medium effect size (d = 0.673). According to the quantitative analyses, educational orientation as a motive was thus considered more important by non-working students compared to working students.

The difference in mean pupil orientation for graduated working students (M = 3.385, SD = 0.843) and non-working students (M = 4.222, SD = 0.581) was also significant: t (16.316) = 3.311, *p* < 0.01, with a large effect size (d = 1.273). This difference, in which non-working students attached more importance to pupil orientation compared to working students, was showed in Figure 7.

The difference in mean idealism for graduated working students (M = 3.139, SD = 1.387) and non-working students (M = 3.911, SD = 0.752) was slightly significant at the exploratory level: t (14.628) = 1.910, *p* < 0.1., with a large effect size (d = 0.809). Working students considered idealism to be less important compared to non-working students. The qualitative data confirm this.

Work perspective, subject orientation, educational orientation, pupil orientation and idealism are less important motives for working students compared to non-working students to opt for teaching. For the motives of educational orientation, pupil orientation and idealism, it is possible that working students experience teacher education as too demanding because they combine it with other work and possibly a family. Due to this negative experience, it is possible that their motivation regarding educational orientation, pupil orientation and idealism is of less importance compared to student teachers who are not working students.

When comparing the started working students with the graduated working students, a number of qualitative differences stood out. Started working students considered ed-

ucational orientation less important than graduated working students. Work dynamic also increased from started working students to graduated working students. This can be explained by the hands-on experience the student teachers have had. After teacher education, the students have a better idea of what the job exactly entails. This can ensure that the dynamic element of the teaching profession becomes more important. Idealism, on the other hand, declined from started working students to graduated working students. Often, student teachers start their education with a romanticized vision of the teaching profession, especially related to pupil orientation. After teacher education, their perspective is often more realistic, which causes the motivation regarding idealism to decline.

## 5. Discussion

This research examined student teachers' motivation to become a teacher and how this evolves throughout their education. First, motives were examined generically, then they were broken down by gender and by type of track. Compared to the period between the start of teacher education and its completion, it appeared that student teachers attached more importance to subject orientation and educational orientation. Pupil orientation, on the other hand, declined throughout the program. The results show a shift from romanticized pupil orientation among started students to a more realistic educational orientation among graduated students. This resembles results of Sutherland et al. [46] on changes in professional identity. They also refer to a romanticized view on teaching which beginning teachers can have, for instance related to the strong ties they hope to build with their pupils, which is changing with experience.

Broken down by gender, the following findings emerged. Work dynamic as a motive to work in education was generally considered equally important by women and men. This result is in line with existing literature, which states that men and women do not differ in their choice of performing challenging and dynamic tasks in a job [47]. Combination possibilities were more important for men than for women. Literature showed that most men are guided by extrinsic job characteristics, such as combination possibilities [48]. This could explain why men attach more importance to combination possibilities than women.

When male starters were compared to graduated men, it was noticeable that work dynamic and work perspective scored higher with graduated men than with male starters. Hands-on experience might explain this. After teacher education, student teachers have a better view on what the job entails. This may cause the dynamic element, as well as the will for certainty in the teaching profession, to become more important. The study by Siongers, Vangoidsenhoven, Kavadias and Glorieux [49] confirms this. Subject orientation, on the other hand, was less important among male graduates compared to men who had just started their education.

Specifically related to students with a 'teacher in training' (TiT) statute, graduate TiT-students considered the motive idealism more important than non-TiT-students. This is because TiT-students learn in practice, which can play a crucial role in enhancing motivation about idealism [50]. The motives of pupil orientation and work perspective declined among TiT-students throughout teacher education. The result was in line with the survival phase that beginning teachers go through. According to Moir [51], teachers in this phase switch to autopilot and find little time to focus on matters besides teaching, such as pupil orientation. Possible causes are workload, lack of cooperation between the training school and the teacher education college or lack of initial guidance. For the work perspective, an explanation was sought for the stress experienced by regular students when looking for a first job, which is not experienced by TiT-students as they already have a job [52,53].

Specifically related to working students, work perspective appeared to be more important compared to non-working students. A possible explanation for this could be that these students already have a job and therefore experience less stress in finding one [52,53]. Subject orientation was more important for non-working students. This finding is contrary to expectations, since working students already have experience in the professional field and can therefore share their expertise with students. Research shows that side entrants,

and therefore also working students, consider the teaching profession as one that requires knowledge and expertise [22]. This perception leads to expect that the more the side entrants believe that a teacher must be an expert in the field and they possess this quality, the more motivated they are to become teachers [38]. For working students, from all their experience, subject orientation may be so self-evident that they no longer think about it in the first place.

Work perspective, subject orientation, educational orientation, pupil orientation and idealism were less important motives for working students compared to non-working students. These figures are surprising since previous research [31] observed the opposite. In Boone's [31] study, student teacher commitment increased as they progressed through teacher education. For the motives of educational orientation, pupil orientation and idealism, it is possible that working students experience the study program as too demanding because they combine teacher education with work and often also a family [54]. Due to this negative experience, there is a chance that their motivation regarding educational orientation, pupil orientation and idealism is therefore of less importance compared to non-working students.

When started working students were compared to graduated working students, some differences stood out. Educational orientation and work dynamic increased in importance. This could be explained by hands-on experience. After teacher education, the students have a better idea of what the job entails. This can make the dynamic element of the teaching profession more important. The study by Siongers, Vangoidsenhoven, Kavadias and Glorieux [49] confirms this. Idealism, on the other hand, declined in importance. Student teachers often start their education with a romantic image of the teaching profession. When graduating, after different experiences in training schools, their perspective is often more realistic [51,55].

These insights into evolving motives among student teachers have a number of possible implications for the organization of teacher education. It is clear that when student teachers have positive hands-on experiences, their motivation to become a teacher may increase [56]. Teacher education can therefore put extra emphasis on positive hands-on experiences by making a well-considered selection of internship schools and mentors. At the same time, teacher education can offer even more space and time for student teachers to dedicate to internships, by keeping other activities such as deadlines and teaching moments during internships to a minimum.

When it comes to internship experiences, it is not only a question of quantity but also of quality. TiT-students learn in practice. With this group of student teachers, it is therefore again important to focus on experiencing positive practice moments. Possible causes of negative hands-on experiences are workload, lack of cooperation between the teacher training school and the teacher education program or lack of initial guidance. Teacher education programs can therefore focus on "education together" [57]. In concrete terms, this means that teacher education colleges and schools work closely together so TiT students experience little transition between teacher education and school. This is done with the help of an on-the-job education program in which learning, training, coaching and assessment are central [58].

More specific in the area of coaching and supervision during internships, teacher education can make further efforts. Usually, an internship supervisor from the university visits the internship once. Based on this snapshot, student teachers are largely assessed. This snapshot can be extended by scheduling several conversations with internship supervisors. During such meetings, working points and difficulties experienced by student teachers during the internships can be dealt with concretely. In this way, the university focuses even more on coaching and not mainly on assessment, which is in line with the results of the review study of Clarke, Triggs and Nielsen [59] on success conditions for cooperating teacher participation in teacher education.

For the motives 'educational orientation', 'pupil orientation' and 'idealism', it is possible that working students experience teacher education as too demanding because

they combine it with other work and often also a family [54]. Most teacher education programs already place great emphasis on flexibility in order to meet the pressure from working students [60]. It will be important, however, not to let the high flexibility of teacher education come at the expense of transparency and consistent structure. Transparency, on one hand, refers to the content of course units (focus on practice or theory, lectures or external assignments, etc.) and the workload that each subject entail. Consistent structure, on the other hand, refers to clarity regarding the entire curriculum (alignment between study guides and individual courses, alignment between subcourses, etc.).

The results of this study should be interpreted in the light of some limitations. First, we were not able to pair the answers of the start survey with the answers of the survey at the end of teacher education (while graduating). Therefore, this research indicates a general trend and not an evolution between a pre- and posttest. Future research can better substantiate evolutions if started students and graduated students are linked to each other. Secondly, our data, as a consequence of a ceiling effect, were slightly skewed. However, we used parametric *t*-tests based in the assumption of normality, to analyze our data. We based this choice on the arguments of scholars who refer to consequences of the central limit theorem, which means that sufficiently large samples which are slightly non normally distributed would be sufficiently robust for *t*-tests and that there would be a trade-off between non-parametric tests which would be less sensitive and therefore are keen to miss effects and the use of parametric tests which are more sensitive [61,62]. In future research, it seems necessary to avoid these ceiling effects by using an extended scale in the Likert items. Further, normally distributed data, linked to an even larger number of respondents, would better suit the statistical conditions for future research.

**Author Contributions:** All authors contributed substantially to the work reported. Conceptualization: E.T. and W.S.; methodology: W.S.; software: K.P., H.V.L. and E.T.; validation, formal analysis, K.P., H.V.L., E.T. and W.S.; investigation, K.P., H.V.L. and E.T.; resources: W.S.; data curation, K.P., H.V.L. and E.T.; writing—original draft preparation, K.P., H.V.L., E.T. and W.S.; writing—review and editing, E.T. and W.S.; supervision: E.T. and W.S.; project administration: E.T. All authors have read and agreed to the published version of the manuscript.

**Funding:** This research received no external funding.

**Institutional Review Board Statement:** The study did not require ethical approval.

**Informed Consent Statement:** Informed consent was obtained from all subjects involved in the study.

**Data Availability Statement:** All data can be obtained from the corresponding author.

**Conflicts of Interest:** The authors declare no conflict of interest.

## Appendix A

**Table A1.** Variable statement.

| Variable | Statements |
| --- | --- |
| Subject orientation | I like to motivate students for my subject, I want to stimulate the love for my subject, I want to teach students something, I want to make students experience pleasure in learning, I want to help students understand my subject, I want to stimulate students to deepen their understanding of my subject, I want to make my subject more interesting. |
| Work perspective | I want to work in education because it offers great job security, I don't see any other job opportunities for me, I can probably find work in education, teaching is a logical consequence of my previous choice of study. |
| Work dynamic | I want a dynamic job, I want a job with a lot of variety, I want a job where I don't have to sit still all the time, I want a lot of freedom in my job, my job should keep challenging me. |

Table A1. *Cont.*

| Variable | Statements |
|---|---|
| Educational orientation | I like leading a group of students, I enjoy teaching, I like standing in front of an audience, I like chatting, teaching suits me. |
| Pupil orientation | I want to help pupils in their choice of study or profession, I want to help pupils, I want to give them opportunities, I want to show them what they can do, I want to prepare them for further studies or a profession, I want to help pupils who have difficulties in learning. |
| Idealism | I want to improve education, I want to contribute to society through education, I want to help shape the next generation, I want to improve the world a little, I want to stir up the idealism of students. |
| Combination possibilities | I like to have good working hours, I like to work close to home, I like to be able to combine my job with a family, I like a lot of holidays. |
| Gender | Gender is broken down into male, female and X. |
| Working student | Students who combine work and study. Within this statute, it is not a requirement to have a job in education. Working students can also be employed in other sectors. |
| TiT-student | Teacher-in-training: someone who is already practicing as a teacher, but who is simultaneously taking teacher training courses. TiT-students follow the theoretical subjects of teacher education together with regular students, but are (partially) valorized for the practical components. TiT-students can carry out internships as part of their job but are still supported in this by teacher training. |

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
