# Peer review of "Motives of Student Teachers in Academic Teacher Education for Secondary Education: Research in Flanders (Belgium) on the Motivation to Become and to Remain a Teacher"

_education, doi:10.3390/educsci12100724_

Round 1

Reviewer 1 Report

General remarks:

The overall manuscript contains relevant information for existing literature albeit presented in a sloppy, incomplete way. Your current version is unsuitable for publication. I would strongly recommend using peer review before you submit a manuscript. More attention need to be focused on coherence and consistency throughout your work. I would also suggest to search for a few best-practice manuscripts in your field to examine and learn from. I was not impressed by the quality of the manuscript. 

Specific remarks:

p.1.4                “shortage of teachers” à In primary education? Secondary education? Higher education? You need to specify this.  

p.1.6                “whether or not…” = redundant. You can nuance this in your main text; not in the abstract. Especially if you in row 8/9 focus on “to start teaching”. This has to be clarified. To start the education, or to finish the education and to actually start teaching.

p.1.6                You spell it here “student-teacher” but later on (i.e., p.1.41) as “student teachers”.

p.1.10/11         This is not a conclusion from your research. Please adjust.

p.1.16              Where? This has to be more specific. In addition, “can be felt in practice” needs to be specified as well.

p.1.27              Avoid adding “etc.” to your summary.

p.1.21/32         I have the feeling you research is situated in Flanders. If so, add this to the title. This is a rather specific context and, in terms of implementation elsewhere, this information is crucial for other researchers.

p.1.37–39        Adding “whether or not” to a research question is a no-go. Please rephrase accordingly.

p.2.55              “van” = “Van” (it needs to be capitalized). Change it in row 62 on this page as well (compare it to p.3.100).

p.2.52/53         Play an important role in what?

p.2.70/76         “In addition” and “In addition to the above motives” à repetition.

p.2                   The theoretical framework seems fragmented. It is incoherent. Can you aim for more coherence (e.g., by inserting signaling words, by repeating [parts of] sentences, and/or by aiming for a logical order of topics). The information is not presented in a logical order.

p.2.64              General statement. There is probably overlap in motivation.

p.2/3                That paragraph is suitable as an introductory paragraph about motives. You can add the additional research to support certain claims. You state in p.3.122 the following: “very personal and depends on the context” à Avoid intensifiers such as “very” + the latter part of that sentence is a direct link to the contextual factors, that you earlier describe as the political, social, and cultural context. Moreover, I would explicitly state to which factor your example refers to. Not every reader is familiar with the political context (e.g. job security). And especially after you state that “This framework therefore forms the starting point for theoretically examining how the motives differ between student-teachers and how they subsequently evolve”. This should be discussed first.

p.3                   If you are using APA 7: delete the bullet points. You can label them (a), (b), etc. If you need to refer to certain motives, it is clear to what you are referring.

p.3.128            Why is this evolution relevant for your manuscript? For this paragraph: I miss the relevance for your story (it feels like extra information that is additional but not necessarily required to understand your work).

p.4.143            I do not understand why you add this here. It is again communicated separately from the rest.

p.4.168            If you change the order in which you present the information, this section will follow logically. In addition, like I mentioned before, I would suggest to add the Flanders context to your title.

p.4.169            Side entrants = you mean students who do not follow the nominal route? I would suggest to move this information to the discussion—as an extension of your own findings. It is a confusing paragraph after the previous sections.

p.5.190–192    And, therefore, this piece of information is better placed in the discussion. It is redundant for the theoretical framework.

p.5                   So you have one main research question and four sub-questions? If you, communicate this. Moreover, a research question has never a “or not” construction (therefore, that part is redundant). In addition, I do not understand your question b. Please clarify. I’ve noticed that your RQs match the different sections in your theoretical framework. I would suggest to implement sub-questions d—if deemed relevant—in the other sub-questions. That distinction is relevant for every sub-question.

p.5.208            Teacher training? Or study?

p.5.209            Group of started students? Do you mean starting students?

p.Table 1        Pose each statement on a new row. In addition, this is suitable for the Appendix. You can communicate one example per theme in the main text. The title of the second column is “Definition”. This is incorrect. It is not a definition nor description. These are the items/statements for that variable. Please adjust the title of the table and the title of the column.

p.5.204            Compare to p.6.230 (plural versus singular form). Keep it consistent.

p.Table 2        You normally do not add the number of participants to the title. Go over the APA 7 rules for suitable guidelines concerning titles for tables. In addition, you need to place the “N” in italics. You total number of participants is 232 + 49 = 281 (= N). The separate numbers require to write the small letter n because it is not your complete sample.  Furthermore, I would replace the comma with a period to meet internal standards (the comma means something else in several countries). To avoid confusion, this has to be changed. In your table you use decimal numbers inconsistently. Write down what you will display with one decimal number (and what with, for example, three decimal numbers). Due to displaying two different overviews, I would separate the two parts. The space on the page can be use more efficiently and, more importantly, it provides more overview for the reader.

p.7.241–248    This information belongs under a section named as “Data analysis”.

p.7.247/248     This requires a scientific source. Moreover, in row 250 you label the T-test as “quantitative”. This word is redundant. A T-test is by definition quantitative.

p.7.250–257    Go over this section to determine what needs to be in the different sections of the methodology. Similar to your theoretical framework, the information is there but it is misplaced and/or there is a lack of coherence.

Table 3           Title needs to be revised. Check the APA manual for the parameters that have to be placed in italics (N, n, t, df, p, etc.). M and SD also need to be placed in italics. Go over the remainder of your manuscript to apply this consistently.

p.8.264            You refer to a Graph 1 whereas the figure is labeled as Figure 1. This has to be consistent.

p.9.275–278    Insert more quotes or qualitative materials to indicate the importance of those dimensions. The quote you insert is not illustrating your point. Please adjust this also for the other dimensions. Qualitative research gives rich information and, at this point, you do not make sufficient use of that.

p.9                   Insert a comma or semi-colon between the M and SD value on this page.

p.9.309            “It amounted to a medium effect” à Odd. Please rephrase. You can look up a few best-practice examples that you can use as a guideline.

p.9.303–306    This information belongs in the discussion. You need to (only) present your results here.

p.9.314–316    This information does not belong here (it belongs in the discussion).

p.10                 What is category X?

p.results          Go over the results and check what are your results and what belongs in the discussion. If you compare it to existing research, move it to the discussion.

p.17.480          “to look at” à to examine.

p.17.485/486   Romanticized pupil orientation?

p.17.507          Different way of writing for teacher in training (compare this to the way it is written in the theoretical framework).

p.discussion    Decent with a few extra spaces every now and then (p.17.523).

p.references    Need more attention, especially if it is APA 7. Issues: hyphen versus end ash, text in italics, capital letter use.

Reviewer 2 Report

Thank you for doing this study. Certain major amendments must be made before publication. 

1.      The title needs modification; (Motives of student-teachers. research on the motivation to become and to remain a teacher.). The current title is a bit confusing; you used a full stop after the first part of the title. Usually, we use a colon to illustrate or emphasize the preceding sentence. What is the level of the student teachers? School or tertiary? You may add a level, context etc.

2.      An abstract should include the following sections: background/motivation, aim, methodology, principal findings, and conclusion/significance. Please rewrite your abstract after revisions. The current form of the abstract is not well clear.

3.      Your general research question is: What motivates student-teachers to become teachers or not (within the teacher education program of the University of Antwerp)? This question has two directions, Yes and NO. Usually, we emphases on one direction and after the results, we can observe the second one. Moreover, if we develop a hypothesis, one would be a null hypothesis, and another would be an alternative hypothesis. So please re-think and change your research question.

Later, Line 194: You also mentioned RQ; treat as suggested before. It would be better to develop hypotheses.

4.      Teacher education, teacher training, teacher education program: are these terms interchangeable? Or do they have different meanings? Please be consistent. Alternatively, you may add definitions of these terms.

5.      I just found that under the data description, you mentioned, ‘The group of Started students was compared to the group of graduated students.’ If you are comparing two groups, your study would be from a different angle, and your research questions should be related to the comparison.

6.      The method section has serious issues; how did you recruit the sample? Which instrument did you use? Was it validated or developed? Which sampling technique did you use? Explain all steps with sub-headings.

Please add the study context, as you are doing this study specifically at the local university, so let your reader know about the university and program. Moreover, if you are making a comparison, establish your viewpoint in the introduction. In another case, just focus on the factors that motivated trainee teachers to remain in the profession.

Reviewer 3 Report

The research seems up-to-date and interesting in terms of its subject. Although the research title is not too long, it is suitable for the purpose. But it is not appropriate to use the words student-teachers here. In the literature, mostly "Preservice" or "Prospective" teachers are used. Similarly, it should be corrected in the article.

It was also bad that the abstract of this research was not in the structured abstract form. In other words, it would not be good not to write a summary including the purpose, method, data collection tool and results in the summary. However, the research findings should also be briefly mentioned in the abstract.

The introduction of the research is appropriate in terms of literature. The introduction part of the research is also sufficient in terms of subject area.

 The bibliographies used are up-to-date. Therefore, the use of new bibliography in the introduction and discussion sections of the research has enriched the research.

Purpose and sub-objectives were written in line with the findings.

The research method was not written. Survey method was used in the research. But in the research, it would be better to write the research design for this study. No information was given about the validity and reliability of the data collection tool. In addition, parametric analyzes were made and no information was given about the distribution of the data.

The number of participants in the study is different and incorrectly written in some tables, and there is confusion.

It is stated as "3.1 Description of the data" but actually this title should be "Participants".

In Table 2, decimals should be separated by periods, not commas.

Care was taken to write the tables used in the research in the form of APA6 standard.

Writing the results of the research in the form of frequencies, excluding parametric or non-parametric tests, calls into question the reliability of the research.

Independent t-test was used in the study. But paired sample t-test could also be used. If the same people were followed, a comparison between before and after graduation could be made.

The discussion section was written in the research, but the conclusion and recommendations section were not written separately. It would be more meaningful to include student opinions and suggestions written accordingly to the research results.

Round 2

Reviewer 1 Report

Hyphen in the reference list needs to be replaced by an en dash. 

Reviewer 2 Report

Dear authors,

Thank you for the revisions and the response letter. Upon reviewing the revised version of the manuscript, I can see that you have made substantial changes. Best wishes.  

Reviewer 3 Report

The authors improved the work by making acceptable corrections to the article. However, there are still some points that need to be corrected.

"variables were not perfectly distributed when the t-tests 667

were carried out because of a ceiling effect in the Likert scale items. Because the number 668 of said was greater than 100, conducting a t-test was still sufficiently robust" was said. But which researcher said this? Reference is needed. Or you can divide the skewness value by its error. It is more convenient way .

The second thing is that "the questionnaire was expanded to include a section on personal data and questions on the organization of the course. This was extensively tested with the same target group and then adjusted. In 2020-2021, the modified written questionnaire was extensively tested and validated". Saying "tested and validated" is not enough. Authors should give statistical values ​​for these tests. Or if it has been published by authors in research, then its reference can be written here.
